# Generation and Identification of the Number of Copies of Exogenous Genes and the T-DNA Insertion Site in SCN-Resistance Transformation Event ZHs1-2

**DOI:** 10.3390/ijms23126849

**Published:** 2022-06-20

**Authors:** Guixiang Tang, Xuanbo Zhong, Wei Hong, Jianfei Li, Yue Shu, Lulu Liu

**Affiliations:** Zhejiang Provincial Key Laboratory of Crop Genetic Resources, Institute of Crop Science, Department of Agronomy, Zhejiang University, Hangzhou 310058, China; 21716135@zju.edu.cn (X.Z.); hongwei1101@foxmail.com (W.H.); 22016146@zju.edu.cn (J.L.); 22016127@zju.edu.cn (Y.S.); lll9012@126.com (L.L.)

**Keywords:** *Hs1^pro−1^* gene, transformation event, SCN, number of gene copies, T-DNA insertion site

## Abstract

Soybean cyst nematode (SCN, *Heterodera glycines* Ichinohe) causes an estimated economic loss of about USD 3 billion each year in soybean (*Glycine max* L.) production worldwide. Overexpression of resistance genes against SCN provides a powerful approach to develop SCN resistance cultivars in soybean. The clarification of molecular characterization in transformation events is a prerequisite for ecological risk assessment, food safety, and commercial release of genetically modified crops. Here, we generated transgenic events harboring the BCN (beet cyst nematode) resistance *Hs1^pro−1^* gene using the *Agrobacterium*-mediated method in soybean, evaluated their resistance to SCN infection, and clarified the molecular characterization of one of the transformation events. Five independent and stable inheritable transformation events were generated by an *Agrobacterium*-mediated transformation method. SCN resistance tests showed the average number of developed females per plant and female index (FI) in T4 ZHs1-1, ZHs1-2, ZHs1-3, ZHs1-4, and ZHs1-5 transformation events were significantly lower than that in the nontransgenic control. Among these, the ZHs1-2 transformation event had the lowest number of developed females per plant and FI. Southern hybridization showed the exogenous target *Hs1^pro−1^* gene was inserted in one copy and the *Bar* gene was inserted two copies in the ZHs1-2 transformation event. The exogenous T-DNA fragment was integrated in the reverse position of Chr02: 5351566–5231578 (mainly the *Bar* gene expression cassette) and in the forward position of Chr03: 17083358–17083400 (intact T-DNA, including *Hs1^pro−1^* and *Bar* gene expression cassette) using a whole genome sequencing method (WGS). The results of WGS method and Southern hybridization were consistent. All the functional elements of exogenous T-DNA fragments were verified by PCR using specific primer pairs in the T5 and T6 ZHs1-2 transformation events. These results demonstrated that the overexpression of *Hs1^pro−1^* gene enhanced SCN resistance, and provide an important reference for the biosafety assessment and the labeling detection in transformation event ZHs1-2.

## 1. Introduction

Soybean plays an important role in China’s economy; China is currently the world’s largest consumer and importer, and the number of imported soybeans is increasing annually [1]. Soybean cyst nematode (SCN), *Heterodera glycines*, is an important worldwide parasite that threatens major soybean production areas, causing an average yield reduction of about 10% to 30%, and in severe cases, 60% to 70%, or even no yield [2]. In general, it causes annual global losses of about USD 3 billion in soybean production [3]. Yield losses caused by SCN are often underestimated because the nematode can be present in the field without causing obvious aboveground symptoms. The SCN is a soil-borne, sedentary, and obligated nematode that parasitizes soybean roots. The developed females (cysts) can survive in the soil for 10 years or more without a host [4]. The cysts hatch to develop the second-stage juvenile under suitable conditions to invade the soybean roots, and complete multiple generations in a single season so that the SCN population density is increased quickly in the field [5]. This results in decreasing the effect of traditional methods against the SCN in the field; for example, using nematicides [5], crop rotation with nonhost crops [2], and growing resistant cultivars [6].

Planting SCN-resistant cultivars is still the most environmentally and economically friendly method to control SCNs. Around 90% of SCN-resistant cultivars currently bred in the central US are derived from the soybean germplasm PI8878, whichhas become noneffective against SCNs due to SCN populations overcoming their resistance to PI8878, and the female index (FI) has increased [7]. So, it is a research priority for controlling SCN worldwide to screen resistant germplasms and discover the high-quality resistant genes under the limited number of naturally resistant cultivars facing failure and depletion. Many scientists have performed a great deal of research on SCN resistance, from plant resistance genes to the nematode gene itself. Expression of *Bacillus thuringiensis* (*Bt*) delta-endotoxin (*Cry14Ab*) in soybean could significantly reduce the number of SCNs compared to control plants 30 d after inoculation due to Cry14Ab directly damaging the intestines of the nematodes [7]. Overexpression of kinase-dead variants of five highly connected syncytium hub genes significantly enhanced soybean resistance to SCNs [8]. Silencing three essential *H. glycines* genes in soybean; i.e., *Hg-rps23*, *Hg-snb1*, and *Hg-cpn1*, enhanced broad-spectrum SCN resistance [9]. Until now, two SCN resistance loci, including the *rhg1-a* (from Peking) and the *rhg1-b* (from PI 88788), have been identified [10]. The unique SCN-resistance candidate gene *GmSHMT*, which is located in the *Rhg*4 locus, was cloned [11]. The *Hs1^pro−1^* gene was the first beet cyst nematode (BCN) resistance gene cloned from the sugar beet translocation line, and it belonged to a transmembrane class protein with an LRR-TM (leucine-rich repeat-transmembrane) structure [12]. Overexpression of the full-length *Hs1^pro−1^* gene coding sequence in *Arabidopsis* root hairs [13], soybean [14], and rapeseed [15] exhibited BCN and SCN resistance. The *Hs1^pro−1^* gene for resistance to the beet cyst nematode in sugar beet was expected to have a resistance function to SCNs in soybean. It was found the *GmHs1^pro−1^* gene cloned from the SCN-resistant cultivar *Wenfeng 7* was similar to the *Hs1^pro−1^* [16]. The relative expression of the *GmHs1^pro−1^* gene was upregulated in SCN-susceptible cultivars after infection with SCNs [17]. In general, generating genetically modified (GM) *Hs1^pro−1^* gene plants might be useful in breeding new SCN-resistant cultivars in soybeans.

GM crops had been planted in a total of 190.4 million hectares of planted area in more than 20 countries around the world by the end of 2019, with GM soybeans having the largest planted area of 91.9 million hectares of GM crops [18]. Although GM crops have been planted worldwide and have brought many benefits to farmers, there is still public debate on these GM crops, and there are worries about the ecological risk and food safety of such crops. Therefore, laws and regulations have been established to ensure GM crops’ safety for human health and environmental risk in various countries [19]. A complete tracking and monitoring system has been developed for a gene-transformation event [19]. A gene-transformation event is the insertion of exogenous genes into a plant genome by molecular bioengineering techniques [20]. The exogenous DNA is randomly inserted into the host genome during the generation of GM plants. The molecular characteristics of GM crops are important indicators to ensure the safety of GM crops [21]. The molecular characteristics include the stability of the T-DNA insert(s), the number of copies of exogenous genes, the insertion sites of T-DNA in the host genome, the flanking sequence of the insertion site, and any unintended DNA sequence from the T-vector. Clarifying the molecular characteristics of GM crops helps to protect the intellectual property rights of the specific transformation event [22]. The number of copies of an exogenous gene is an important factor that affects the expression level and genetic stability of the exogenous inserted genes. The integration of multiple copies of exogenous DNA into one or more chromosomes can reduce gene-expression levels, which may affect the genetic stability of the exogenous gene or lead to gene silencing [23,24]. The ideal transgenic events require a low copy number of the target gene, typically one or two [25]. The number of exogenous gene copies is usually assayed by Southern blot hybridization [26] and qRT-PCR methods [20,27,28]. Southern blot hybridization has the advantages of high accuracy, high specificity, and direct observation of the copy number of the target gene. The T-DNA insertion site indicates where the T-DNA vector is located on the host chromosome. The T-DNA insertion site is a label of each commercially released GM crop, and is also used for screening, identification, and evaluation of the ecological risk assessment of GM crops. In recent years, the whole genome sequence (WGS) has been used to evaluate the T-DNA insertion site [29], and its costs have decreased dramatically.

In this study, overexpression of a sugar beet (*Beta procumbens*) *Hs1^pro−1^* gene (gene accession number: *U79733*) was generated to study the SCN resistance by an *Agrobacterium*- mediated transformation method in soybean. The SCN-resistance assay was conducted on independent transformation events, and such events that had the lowest number of developed females and FI were selected for clarifying the number of exogenous gene copies and T-DNA insertion sites in the host genome. These data suggested the *Hs1^pro−1^* gene plays an important role in SCN resistance, and the identification of the molecular characteristics in transformation event ZHs1-2 could provide an important reference for future ecological risk assessment and cultivar commercial release.

## 2. Results

### 2.1. Generation, Selection, and Inheritances of Transgenic Soybean Lines for Expression of Hs1^pro−1^ and Bar Genes

The T0 ZHs1-2 was acquired by an *Agrobacterium tumefaciens*-mediated transformation method using a seedling cotyledon aged 1–2 d after germinating the soybean cultivar *Tianlong 1* as an explant. The process of *Agrobacterium*-mediated soybean transformation consisted of several stages (Appendix A); i.e., seed sterilization (Appendix A), seed germination (Appendix A), cotyledonary explant isolation and infection (Appendix A) and cocultivating with *Agrobacterium* (Appendix A), shoot induction (Appendix A), shoot elongation (Appendix A), shoot rooting (Appendix A), and plantlet domestication (Appendix A). A total of 69 putative transgenic plantlets were generated among isolated 1388 explants (Table 1); and 39 positive transgenic seedlings of T0 progeny, which were identified by coating leaves with glufosinate (Figure 1A), target gene *Hs1^pro−1^*, screening marker *Bar* gene PCR (Figure 1B), and Quick Bar protein detection (Figure 1C), were generated. The transgenic efficiency varied widely among the different transformation batches, with the highest being 6% and the lowest 0. The average transformation efficiency was 2.81% (Table 1). Among 39 T0 independent transformation events, 5 transformation events were sterile, 11 transformation events failed to survive due to poor management, and only 23 independent transformation events were able to self-pollinate to generate T1 seeds. Segregation occurred in the T1 progeny (Table 2). Among 23 T1 independent transformation events, only 9 transformation events were stably inherited into the T2 progeny. The remaining 14 T1 independent transformation events were not positive for the desired target genes (Table 2). In addition, a chi-squared test showed that the segregation ratio of the exogenous genes in 152, 175, 187, 188, and 196 independent transformation events was based on Mendelian genetics (Table 2). The five independent transformation events were named ZHs1-1, ZHs1-2, ZHs1-3, ZHs1-4, and ZHs1-5 to investigate the SCN-resistance assay. Each transgenic progeny was developed by self-pollination; the assay parameters of each progeny are shown in Appendix A.

### 2.2. SCN Resistance in Transformation Events

The T4 independent soybean transformation events (ZHs1-1, ZHs1-2, ZHs1-3, ZHs1-4, and ZHs1-5) were used to evaluate race 4 of SCN resistance in a greenhouse. Race 4 of SCN mainly occurred, and it damaged the soybean production in the Huanghuai-Hai region of China. The number of developed females per plant was counted 35 days after SCN inoculation. There was a significant difference in the number of developed females between transformation events and controls, including NT (nontransgenic control) and SCN-susceptible cultivar ‘*Lee 68*’ (Figure 2A). The number of developed females per plant in T4 ZHs1-1, ZHs1-2, ZHs1-3, ZHs1-4, and ZHs1-5 was 31.3, 25.8, 42.4, 33.1, and 44.5, respectively. Conversely, the number of developed females per plant in the NT and susceptible cultivar ‘*Lee 68*’ was 80 and 67.8, respectively. The number of developed females per plant in transformation events decreased by an average of 44.6 and 32.4, a decrease of 126% and 91.42%, respectively, compared to the NT and susceptible cultivar ‘*Lee 68*’. Among five transformation events, the number of developed females of ZHs1-2 had the lowest (25.8), with a 54.7 and 42.1 reduction compared to the NT and susceptible cultivar ‘*Lee 86*’. Similarly, the FI (female index) in T4 ZHs1-1, ZHs1-2, ZHs1-3, ZHs1-4, and ZHs1-5 was 46.2, 38.0, 62.5, 48.8, and 65.6, respectively. It was lower than that of NT (117.9) (Figure 2B). The FI in transformation events decreased by an average of 74.9, a decrease of 63.5% compared with the NT. In particular, the FI in ZHs1-2 transformation events had the lowest (38), a decrease of 79.91% compared with the NT. Based on the results, overexpression of *Hs1^pro−1^* gene in soybean could enhance the SCN resistance by reducing the number of developed females per plant and FI. So, the ZHs1-2 transformation event that generated the lowest number of developed females per plant and FI was selected for further molecular characteristics identification.

### 2.3. Copy Number of Exogenous Gene in ZHs1-2 Transformation Event

The result of the Southern hybridization of the exogenous *Hs1^pro−1^* gene is shown in Figure 3A. The location of the probe for the exogenous target gene *Hs1^pro−1^* and the enzymatic sites of the restriction endonucleases *HindIII* and *EcoRV* are shown in Figure 3B. Using *Hind*III restriction enzyme digestion and hybridization with the specific probe, a 10.7 kb band was observed when hybridization was performed with the positive control (Figure 3A, lane P); no band appeared when hybridization was performed with the genomic DNA of NT *Tianlong 1* (Figure 3A lane 1); and a 3.0 kb band that consisted of 2.0 kb of the T-DNA sequence and unknown sequences on either side of the genome was generated when hybridization was performed with the genomic DNA of the ZHs1-2 transformation event (Figure 3A, lane 3). Using *EcoR*V restriction enzyme digestion and hybridization with a specific probe, no band appeared when hybridization was performed with the genomic DNA of NT *Tianlong 1* (Figure 3A lane 2). An approximately 7.0 kb band that included 1.6 kb T-DNA sequences and an unknown-size sequence on the downstream host genome was generated when hybridization was performed with genomic DNA of the ZHs1-2 transformation event (Figure 3A, lane 4). The results indicated that the exogenous target gene *Hs1^pro−1^* was inserted into the host genome as a single copy in the ZHs1-2 transformation event.

Results of the Southern hybridization of the selection marker *Bar* gene are shown in Figure 4A,B. The location of the probe for the *Bar* gene and the enzymatic sites for the restriction endonucleases *Kpn*I and *EcoR*V are shown in Figure 4C. Using *KpnI* restriction enzyme digestion and hybridization with the specific probe, a 10.7 kb band was observed when hybridization was performed with the positive control (Figure 4A, lane P). There was no band appearance when hybridization was performed with the genomic DNA of NT *Tianlong 1* (Figure 4A, lane 1). The bands should have been greater than 3.3 kb, including a 3.3 kb T-DNA sequence and sequences of unknown size on either side of the host genome, when was hybridization performed with the genomic DNA of the ZHs1-2 transformation event. Two bands were accidentally generated, and the band sizes were approximately 3.7 kb and 4.0 kb (Figure 4A lane 2). Using *EcoR*V restriction enzyme digestion and hybridization with the specific probe, a 9.7 kb band was generated when hybridization was performed with the positive control (Figure 4B, lane P). There was no band appearance when hybridization was performed with the genomic DNA of NT *Tianlong1* (Figure 4B, lane 1). Bands of more than 1.6 kb, including the 1.6 kb T-DNA sequence and sequences of unknown size on either side of the host genome, should have been generated when the hybridization was performed with the genomic DNA of the ZHs1-2 transformation event. The results showed that two bands of approximately 5.9 kb and 6.4 kb appeared when hybridization was performed with the genomic DNA of the ZHs1-2 transformation event (Figure 4B, lane 2). The results indicated that there were two copies of the exogenous *Bar* gene inserted in the transformation event ZHs1-2.

### 2.4. T-DNA Insertion Sites of ZHs1-2 Transformation Event

The whole genomic sequencing of the ZHs1-2 transformation event was obtained using second-generation genome resequencing (BGISEQ-500 WGS). The whole genome sequencing of ZHs1-2 was aligned in the Plant GDB database (http://www.plantgdb.org/GmGDB/cgi-bin/blastGDB.pl, accessed on 5 March 2021) using Williams 82 A2 as the reference sequence. According to the physical positions of junction reads, two copies of foreign T-DNA were integrated into the host genome, at the Chr02: 5351566 to 5231578 position in the reverse direction and at the Chr03: 17083358 to 17083400 positions in the forward direction, respectively (Figure 5).

The genes near the T-DNA insertion site within 5 kb upstream and downstream were predicted using the ‘Genome Context’ tool. The Glyma02906560.1 gene was found 4 kb downstream of the T-DNA insertion site in Chr02, but the T-DNA insertion site was far away, which may not have affected its gene function. No known genes were found within 5 kb upstream or downstream of the T-DNA insertion site in Chr03. A comparison of the T-DNA insertion region on both chromosomes revealed that the left and right bounders were not perfectly inserted. The *Bar* and *Hs1^pro−1^* gene-expression cassettes were entirely inserted into the Chr03, but 24 bp of the left border sequence and 144 bp of the right border sequence were missing. The 2390 bp of the T-DNA region including the sequence of position at 6456–8689 (mainly the *Bar* gene-expression cassette) and at 10426–10581 bp was inserted into the Chr02 genome because 30 bp of the left border and 133 bp of the right border sequence were missing. The *Hs1^pro−1^* gene-expression cassette at vector position 8690–10425 bp was completely lost when the T-DNA vector was transferred.

The functional elements of T-DNA region insertion in the T5 and T6 ZHs1-2 transformation events were verified by PCR using specific primer pairs (Table 3). The results showed that each functional element in the T-DNA region could be amplified to obtain the expected target band size (Figure 6A–I). The results showed that the T-DNA region was functionally inserted into the ZHs1-2 transformation events, and could be stably inherited in the T5 and T6 progeny. The ZHs1-2 transformation event could be identified using PCR based on these specific primer pairs.

## 3. Discussion

SCN is one of the most destructive parasites worldwide. There are limitations to SCN-resistance breeding. SCN resistance is controlled by multiple genes, and is also quantitative according to environmental factors [30]. There is an extremely complex host–pathogen interaction between soybean and SCNs, as well as the heterogeneity of the SCN population [30]. Previous studies showed overexpression of *Hs1^pro−1^* gene could enhance BCN resistance in oilseed rape [12,15,31] and soybeans [14]. In this study, the overexpression of *Hs1^pro−1^* gene in soybean plants was generated, and the transgenic plants were evaluated for resistance to race 4 of the SCNs. The results indicated that overexpression of *Hs1^pro−1^* gene in soybean decreased the number of developed females and FI (Figure 2). These results were consistent with previous studies [12,14,15]. The *Hs1^pro−1^* gene was driven by the CaMV 35S promoter in this experiment. The relative expression of the *Hs1^pro−1^* gene after infection with race 4 of the SCNs was not induced from the beginning until 12 days after SCN infection (Figure 7). The mechanism of SCN resistance in transgenic *Hs1^pro−1^* gene plants will be further investigated.

To understand the molecular characteristics in transformation events is a prerequisite for commercial release of GM crops, food security, and environmental-risk assessment. The molecular characteristics of GM crops include the copy number of exogenous genes, integrated location, and flanking sequences, and so on. The copy number of exogenous genes is an important factor that affects the genetic stability and expression level in transgenic progeny. A low copy number (one or 2) of inserted exogenous genes can be well expressed, while a high copy number of the genes leads to unstable expression or even gene silencing [32]. In this study, one copy of the exogenous *Hs1^pro−1^* gene and two copies of the exogenous *Bar* gene were inserted in the transformation event ZHs1-2. The results indicated that two T-DNA cassettes were inserted into the host genome, one intact and the other incomplete. A loss of genes in the T-DNA cassette was observed in transgenic rice after using particle bombardment, and this may be related to the loss of genes during the inheritance of progeny [33]. The loss of T-DNA expression cassette during gene transformation should be studied further. The whole genomic sequencing method was accurately used to identify the insertion site of the ZHs1-2 transformation event. The PCR method was used to verify the flanking sequence, which was consistent with previous studies. Xu et al. (2018) found all the T-DNA insertion sites in three transformation events were successfully obtained by resequencing and using the transgenic vector sequence as a reference for alignment analysis. The traditional PCR-based insertion site identification methods have the disadvantage of being inefficient in analyzing soybean gene-transformation events because soybean is a paleotetraploid crop with nearly 75% of its genes in multiple or complex copies. With the development of high-throughput sequencing technologies, it has become more efficient to use resequencing to accurately identify insertion sites and their paralogous sequences in soybean transformation events [34]. Guo et al. (2016) and Zhong et al. (2018) obtained similar results in the assessment of insertion sites using the WGS method in soybean and oilseed rape. Zhang et al. (2020) reported the full molecular characterization of one new transgenic rice event G6H1 identified via a paired-end sequencing approach and bioinformatics analysis pipelines. These results indicated that high-throughput resequencing could quickly and effectively detect the insertion sites and their flanking sequences. The application of WGS in routine analyses of GM crops will speed up the safety evaluation process of GM crops while ensuring cost-effectiveness and affordability.

The agronomic performance and yield traits of T6 ZHs1-2 transformation events in 2020 and 2021 were also investigated in the field in this study. The results showed that there were no significant differences in plant height, bottom pod height, number of main stem nodes, number of branches, or 100-seed weight between ZHs1-2 transformation events and the NT (Table 4). However, the number of pods and yield per plant were significantly different between ZHs1-2 transformation events and the cultivar *Tianlong 1*. The yield of ZHs1-2 transformation events (1733.5 ± 83.5 kg/hm^2^) was greater than that of the cultivar *Tianlong 1* (1495.4 ± 51.6 kg/hm^2^). In summary, our results demonstrated that the overexpression of *Hs1^pro−1^* gene enhanced SCN resistance in the ZHs1-2 transformation event without negative consequences on the agronomic traits. This suggested that the ZHs1-2 transformation event could be applied in field production and used as a germplasm for SCN-resistance breeding in the future.

## 4. Materials and Methods

### 4.1. Materials

The soybean cultivar *Tianlong 1*, developed and presented by Prof. Xinan Zhou from the Institute of Oilseed Crops Research, Chinese Academy of Agricultural Sciences, Wuhan, China, was used as the transgenic recipient; i.e., the nontransgenic control (NT). The plasmid vector *p*Hs1 (Figure 8) was used for transformation containing the target gene *Hs1^pro−1^* open reading frame of 849 bp. The phosphinothricin acetyltransferase gene (*Bar*) was used as a screening marker gene to show resistance to the herbicide glufosinate. The *Hs1^pro−1^* and *Bar* genes were driven by a CaMv35S constitutive promoter. The position, size, and functions of the vector elements are shown in Table 5. The recombinant binary vectors were transformed into *A. tumefaciens* strain EHA105.

### 4.2. The ZHs1-2 Transformation Events Generation

The ZHs1-2 transformation event was acquired using a *Agrobacterium tumefaciens*-mediated transformation method using mature seed cotyledonary germinated for 1 d as explants [35] after self-pollination of multiple progenies and target-gene identification. T0 transgenic soybean plants were identified using the leaf herbicide tolerance assay, polymerase chain reaction (PCR) targeting the *Hs1^pro−1^* and *Bar* genes using specific gene primers, and Quick Bar protein detection. The primers for target gene PCR detection are shown in Table 6. Quick detection of the Bar protein was performed using the QuickStix^TM^ kit for the *Bar* gene according to the manufacturer’s instructions (Envirlogix Inc., Portland, ME, USA). T0 transgenic plants were self-pollinated to obtain T1 progeny, T1 transgenic plants were self-pollinated to obtain T2 progeny, and so on. T2 independent transformation events were used for cyst nematode resistance identification. T5 and T6 ZHs1-2 were used for the copy number of exogenous genes and T-DNA insertion site analysis.

### 4.3. Resistance to SCN Race 4 Assay

To evaluate SCN resistance of the transformation events, a nematode population of race 4 was used for inoculation assays. SCN race 4 was widely distributed across the Huang-Huai-Hai soybean-growing region in China, and is virulent in most soybean germplasms and cultivars. Soil infected with the SCN race 4 was collected from Shanxi, China, and maintained on the susceptible cultivar Lee under greenhouse conditions. All plants were planted in greenhouse conditions in a plastic cylinder (3 cm × 4 cm × 20 cm) containing ~ 100 SCN eggs/100 g of soil. As controls, the nontransgenic control ‘*Tianlong 1*’ and the susceptible cultivar Lee 68 were also inoculated with the SCN-infected soils. The bioassays for SCN resistance were performed in 7 cylinders for each transformation event and control plant; 2 plants were planted in each cylinder. Greenhouse conditions and the bioassay process were described by Matthews et al. (2016) [36]. At about 35 days after inoculation (dpi), the roots were gently removed from the columns, and the number of developed females was manually counted using a dissecting microscope, as described by Zhong et al. (2019) [15]. The female index (FI) was calculated as follows: Female index = (Nt/Nc) × 100, where Nt is the number of females on roots of transgenic plants and Nc is the number of females present on roots of control Lee 86 plants. Statistically significant differences between the transformation events and the control ‘*Tianlong 1*’ plants were determined using analysis of variance (ANOVA) with *p* < 0.05.

### 4.4. RNA Extraction and qRT-PCR Analysis

Root tissues were sampled 4, 8, 12, and 20 days after SCN infection, frozen quickly in liquid nitrogen, and stored at −80 °C for RNA extraction and qRT-PCR analysis. Total RNA was isolated from tissues of NT and the transgenic event ZHs1-2 using TRIZOL reagent (Invitrogen, Carlsbad, CA, USA) according to the manufacturer’s protocol. Total RNA was quantified with a nanodrop machine. Each sample (200 ng RNA) was reversed-transcribed to the first-strand cDNA using the following PCR mixture (Vazyme Biotech Co. Ltd., Nanjing, China): 4 μL of 5 × Hicrip qRT SuperMix, 4 μL of gDNA wiper, 2 μL of RNA, and 10 μL of RNase-free water. Reverse-transcription PCR was conducted at 50 °C for 15 min. Quantitative real-time PCR was carried out on the CFX 96 Real-Time system (Bio-Rad, Laboratories Inc. US) using ChamQ SYBR qPCR Master Mix (Vazyme Biotech Co. Ltd., Nanjing, China). First-strand cDNA (2 μL each sample) was used for gene-transcript-level analysis with 10 μL 2 × ChamQ SYBR qPCR Master Mix, 0.4 μL each of a pair of gene-specific primers in a final 20 μL volume. The *GmActⅡ* gene served as an internal control, and the *Hs1^pro−1^* gene-specific primer pairs for qRT-PCR were TTGCTGTGATTGGTGGTTCTAC and TTCGCAGTCCGATTCTTCC. The amplification program for the ChamQ SYBR qPCR was performed at 94 °C for 30 s, followed by 35 cycles at 94 °C for 30 s, 52 °C for 30 s, and 72 °C for 50 s. The relative gene expression was calculated using the 2^−ΔΔCt^ method [37].

### 4.5. Exogenous Gene Copy Number Verification

The copy number of *Hs1^pro−1^* and *Bar* genes in the ZHs1-2 transformation events were determined using Southern hybridization (Southern, 1975) [26]. The copy number assay of the *Hs1^pro−1^* gene was performed using two restriction endonucleases. *HindIII* and *EcoRV* were used to digest DNA of the positive control *p*Hs1 plasmid, the nontransgenic control ‘*Tianlong 1*’, and T5 of the ZHs1-2 transformation events. *HindIII* had no restriction endonuclease site in the vector backbone, and had only one restriction endonuclease sites in the T-DNA region, which was located on the left side of the hybridization probe. *EcoRV* had no restriction endonuclease sites in the vector backbone and two restriction endonuclease sites in the T-DNA region, located on the left side of the hybridization probe. *Bar* gene fragment insertion copy number assays were performed using two restriction endonucleases, *KpnI* and *EcoRV*, to digest the DNA of the positive-control pHs1 plasmid, the nontransgenic control ‘*Tianlong 1*’, and the T5 ZHs1-2 genomic DNA, respectively. *EcoRV* had no restriction endonucleases site in the backbone of the vector and two restriction endonuclease sites in the T-DNA region, both located on the right of the hybridization probe. The copy number of each inserted functional element is shown as a specific band after hybridization with the specific probe in the genome. Information on the exogenous gene Southern probe is shown in Table 7.

### 4.6. T-DNA Insertion Site Analysis and Verification

The genomic sequence of the ZHs1-2 transformation event was obtained using second-generation whole genome resequencing (BGISEQ-500 WGS). The sequence containing the vector was extracted and compared with the sequence to analyze the integrity of the inserted T-DNA sequence. Chimeric sequences containing vector sequences and soybean genomic sequences were compared with the soybean reference genome to analyze the insertion sites of exogenous sequences in the soybean genome. Primers were designed for the main functional element sequences (promoter, target gene, and terminator), and PCR was used to detect the integration of T-DNA in the ZHs1-2 transformation events. Information on the primers used is shown in Table 3.

## 5. Conclusions

In summary, the ZHs1-2 transformation event was generated using an *Agrobacterium*-mediated transformation method, and showed the lowest number of developed females per plant and FI after being infected with race 4 of SCN. The number of copies of exogenous genes and T-DNA insertion sites were determined in the transformation event ZHs1-2 using Southern hybridization and whole genomic resequencing. The exogenous target *Hs1^pro−1^* gene was inserted in one copy and the *Bar* gene was inserted two copies in the ZHs1-2 transformation event. The exogenous T-DNA fragment was integrated in the reverse position of Chr02: 5351566–5231578 (mainly the *Bar* gene-expression cassette) and in the forward position of Chr03: 17083358–17083400 (intact T-DNA, including the *Hs1^pro−1^* and *Bar* gene-expression cassette). All the functional elements of the T-DNA region insertion could be stably verified by PCR using specific primer pairs in different progenies. There were no adversely agronomic traits in the transformation event ZHs1-2. Our study provided important information for future ecological risk assessment and cultivar commercial release.

## Figures and Tables

**Figure 1 ijms-23-06849-f001:**
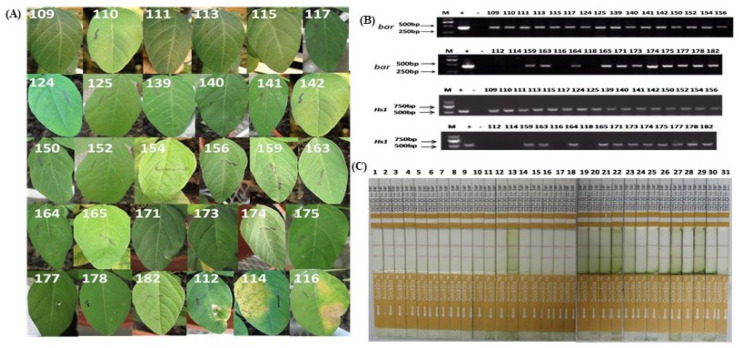
Schematic diagram of putative transgenic plant identification in soybean. (**A**) The leaf herbicide tolerance assay. T0 plants were screened for tolerance to the herbicide Basta by application of a 135 mg/L Basta solution with a cotton swab on the upper surface of the euphylls. Leaf tissue was assessed for herbicide tolerance at 6 or 7 d after herbicide application. (**B**) Polymerase chain reaction (PCR) analysis for target *Hs1^pro−1^* and *Bar* genes using specific gene primers. (**C**) Quick Bar protein detection. Quick detection of the Bar protein was performed using the QuickStix^TM^ kit for *Bar* gene according to the manufacturer’s instructions (Envirlogix Inc., Portland, ME, USA). The presence of a test line (second line) on the membrane strip between the control line (common to all, including the nontransformed control) and the protective tape indicated the expression of the foreign Bar protein in the transgenic plants.

**Figure 2 ijms-23-06849-f002:**
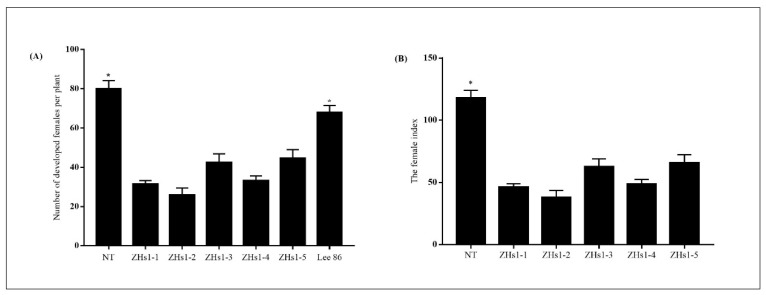
The number of developed females per plant (**A**) and female index (**B**) 35 days after inoculating the SCN race 4 in soybean. NT designates the nontransgenic control ‘*Tianlong 1*’. Lee 86 is the susceptible cultivar of race 4 of SCN. ZHs1-1, ZHs1-2, ZHs1-3, ZHs1-4, and ZHs1-5 represent the T4 independent transformation events. Bars represent the standard errors of the number of developed females per plant and the female index based on 14 independent plants of each transformation event and controls, including NT and Lee 86. The asterisks indicate significant differences between transformation events and NT at a level of 0.05.

**Figure 3 ijms-23-06849-f003:**
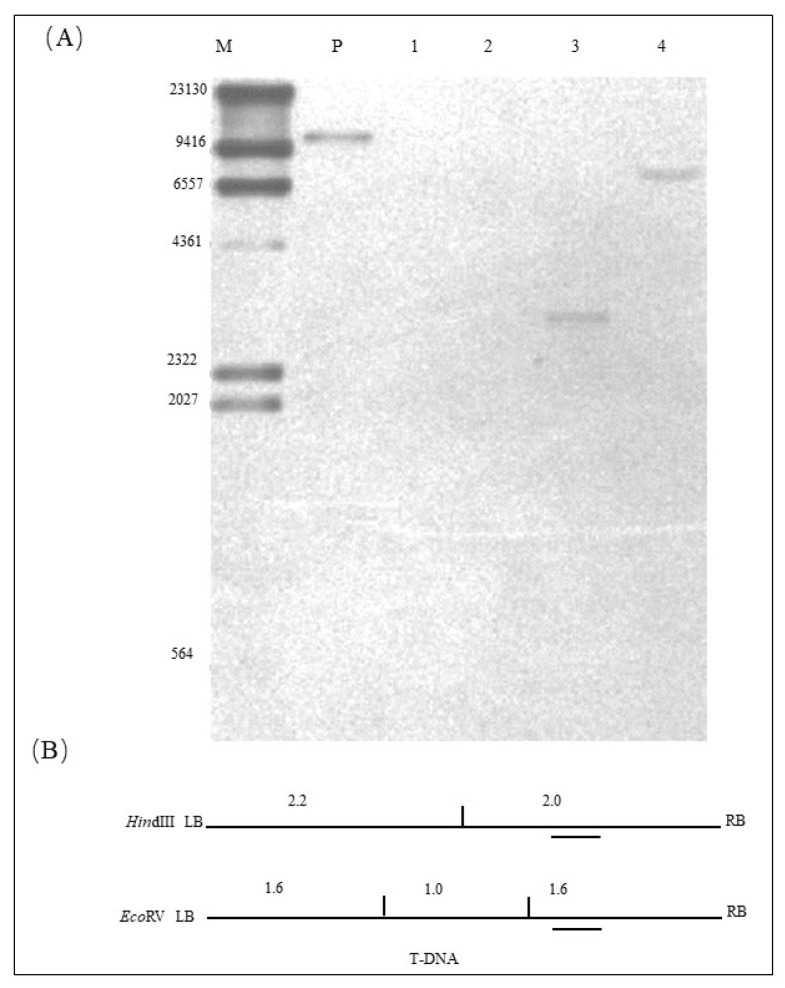
Southern hybridization image of *Hs1^pro−1^* gene (**A**) and a schematic diagram of the restriction enzyme digestion site and the specific probe location of T-DNA (**B**). In (**A**): M, DNA Marker III (bp), DIG-labeled (Roche); P, positive control, *Hs1^pro−1^* gene plasmid DNA; 1, the NT ‘*Tianlong 1*’ genomic DNA digested with *Hind*III restriction enzyme; 2, NT ‘*Tianlong 1*’ genomic DNA digested with *EcoR*V restriction enzyme; 3, T5 ZHs1-2 transformation event genomic DNA digested by the *Hind*III restriction enzyme; 4, T5 ZHs1-2 transformation event genomic DNA digested with *EcoR*V restriction enzyme. In (**B**), the vertical lines indicate the location of the restriction enzyme site digestion and the digested band size (bp); the short dashes indicate the location of the hybridization probe.

**Figure 4 ijms-23-06849-f004:**
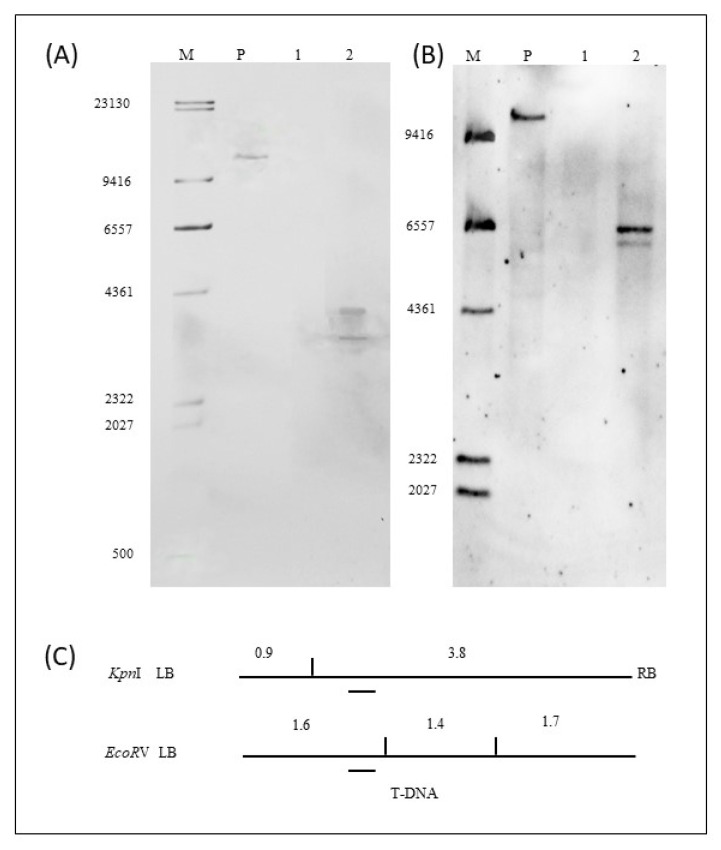
Southern hybridization image of *Bar* gene digested by *Kpn*I (**A**) and *Eco**R*V (**B**), and a schematic diagram of restriction enzyme site and probe location of T-DNA (**C**). In (**A**,**B**): image, M, DNA Marker III (bp), DIG-labeled (Roche); P, positive control of *Hs1^pro−1^* gene plasmid DNA; 1, the NT ‘*Tianlong 1*’; 2, T5 ZHs1-2 transformation event genomic DNA. In (**C**), the vertical lines designate the location of the restriction enzyme site digestion and the digested band size (bp); the short dashes indicate the location of the hybridization probe.

**Figure 5 ijms-23-06849-f005:**
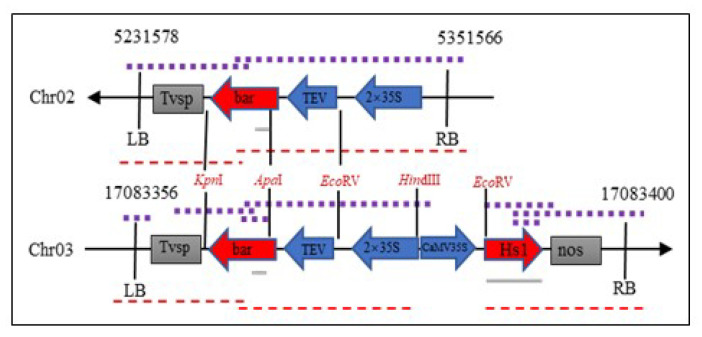
Schematic diagram of a T-DNA insertion sites of the ZHs1-2 transformation event. LB, left border; RB, right border. The purple dotted line represents the location of the PCR amplified product; the red dotted line represents sequencing fragments, and the grey solid line represents the Southern hybridization probe.

**Figure 6 ijms-23-06849-f006:**
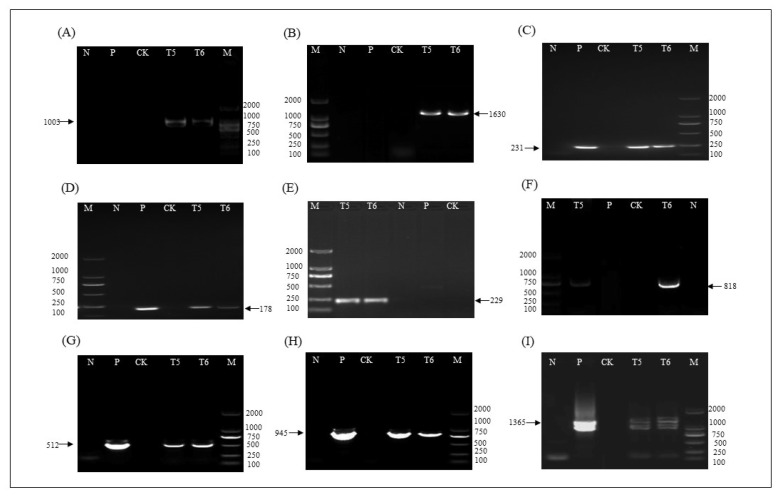
The PCR analysis of each genetic element in T5 and T6 ZHs1-2 transformation events. (**A**) Left border at Chr 02 with the predicted target band size 1003 bp; (**B**) right border at Chr 02 with the predicted target band size 1630 bp; (**C**) *Bar* gene with the predicted target band size 231 bp; (**D**) *Hs1^pro−1^* gene with the predicted target band size 178 bp; (**E**) left border at Chr 03 with the predicted target band size 229 bp; (**F**) right border at Chr 03 with the predicted target band size 818 bp; (**G**) terminator *Tvsp* and *Bar* gene with the predicted target band size 512 bp; (**H**) *Hs1^pro−1^* gene and terminator *nos* gene with the predicted target band size 945 bp; (**I**) *Bar* gene and *2 × 35S* promoter with the predicted target band size 1365 bp. M, DNA Marker (from top to bottom: 2 kb, 1 kb, 750 bp, 500 bp, 250 bp, 100 bp); N, H_2_O; P, positive control pHS1 plasmid DNA; CK, non-transgenic control ‘*Tianlong 1*’; T_5_ and T_6_, T5 and T6 ZHs1-2 transformation events, respectively. The arrows indicate the size of target bands.

**Figure 7 ijms-23-06849-f007:**
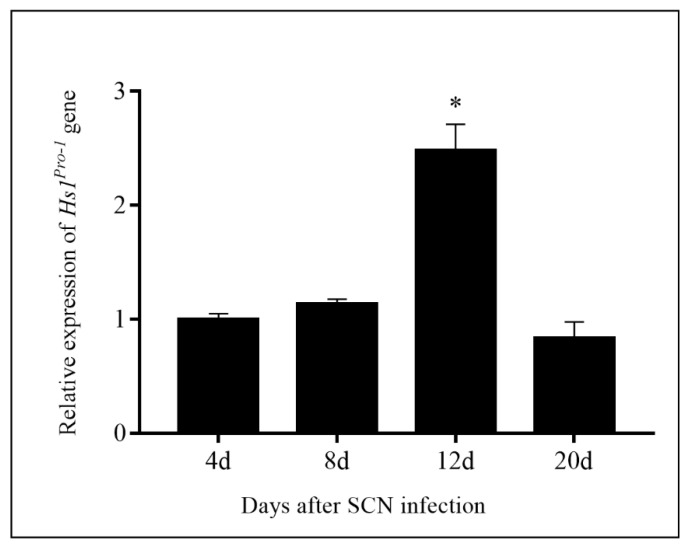
Relative expression of *Hs1^pro−1^* gene in the roots of ZHs1-2 transformation events 4 d, 8 d, 12 d, and 20 d after SCN infection as determined by *q*RT-PCR. Expression was relative to that before SCN infection, the value of which was set as 1. The bars represent the standard error based on three independent biological replicates. Asterisks indicate significant differences at a level of 0.05 as determined by Duncan’s *t*-test.

**Figure 8 ijms-23-06849-f008:**
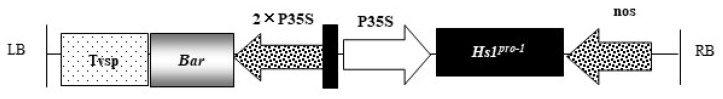
T-DNA region of the expression binary vector *p*Hs1. The *Hs1^pro−1^* and *bar* genes were driven by a CaMV 35S promoter. The *Bar* gene conferred glufosinate tolerance for selection of transgenic plants. LB, left border; RB, right border; Tvsp, soybean storage protein terminator; *Bar*, phosphinothricin acetyltransferase gene; P35S, cauliflower mosaic virus (CaMV) 35S promoter; *Hs1^pro−1^*, gene (gene accession number: U79733) ORF; nos, nopaline synthase terminator.

**Table 1 ijms-23-06849-t001:** T0 putative transgenic plant identification and transformation efficiency.

Experimental Batches	Number of Explants	Number of Rooting Plants	Identification of T0 Putative Transgenic Rooting Plants				Transformation Efficiency %
			Herbicide Tolerance Assay	Quick Bar Protein Assay	PCR for *Hs1* Gene	PCR for *Bar* Gene	
X1	156	0	0	0	0	0	0.00
X2	112	6	4	4	4	4	3.57
X3	180	0	0	0	0	0	0.00
X4	120	8	4	4	4	4	3.33
X5	270	8	4	4	4	4	1.48
L1	150	13	5	5	5	5	3.33
L2	200	20	10	10	10	10	5.00
L3	200	14	12	12	12	12	6.00
Total	1388	69	39	39	39	39	2.81

**Table 2 ijms-23-06849-t002:** The segregation of T1 transformation progeny and chi-squared (χ^2^) test analysis.

Transgenic Plant Code	Number of T1 Seeds	Number of Positive T1 Seeds	T1 Segregation Ratio (Positive:Negative)	χ^2^ Value	Significant χ^2^ Value
150	35	3	1:10	78.87	No
152	40	32	4:1	0.3	Yes
154	4	0	0		
156	10	4	2:3	4.8	No
159	11	0	0		
163	33	18	6:5	6.31	No
164	11	0	0		
171	2	0	0		
173	6	0	0		
174	16	0	0		
175	5	2	2:3	1.67	Yes
178	25	0	0		
186	5	0	0		
187	4	3	3:1	0.33	Yes
188	3	1	1:2	1	Yes
189	4	0	0		
190	3	0	0		
191	9	0	0		
192	13	6	1:1.67	5.77	No
193	3	0	0		
194	25	0	0		
195	5	0	0		
196	12	10	5:1	0.11	Yes

**Table 3 ijms-23-06849-t003:** Name of primers, primer sequence (5′-3′), location in vector (bp), amplified size (bp), and usage for inserted sequence verification.

Name of Primers	Primer Sequence (5′-3′)	Location in Vector (bp)	Amplified Size (bp)	Usage
LC0266	CATTTCACCCTAGTATAACCC	genomic	1003	Detection of Chr02 left boundary sequence, Tvsp, and *Bar* gene
LC0264	CTGGCATGACGTGGGTTT	7314–7331		
LC0084	CCAGAAACCCACGTCATGCCA	7310–7330	1630	Detection of Chr02 right boundary sequence, 2 × 35S, and *Bar* gene
LC0270	ATTGGAGTGGCAAAGGGA	genomic		
LC0236	CAGGTGGGTGTAGAGCGTG	7472–7490	231	Detection of *Bar* gene
LC0237	GTCAACTTCCGTACCGAGCC	7683–7702		
LC0222	TTGCTGTGATTGGTGGTTCTAC	9816–9837	178	Detection of *Hs1pro^−1^* gene
LC0223	TTCGCAGTCCGATTCTTCC	9975–9993		
LC0276	TTGGGGAAGGAAAAGAAT	genomic	229	Detection of Chr03 left boundary sequence
LC0277	TTGTCTAAGCGTCAATT	6453–6469		
LC0222	TTGCTGTGATTGGTGGTTCTAC	9816–9837	818	Detection of Chr03 left boundary sequence, *nos*, and *Hs1pro^−1^* gene
LC0267	CTGCGAGTTGTGAGTTGTGGT	genomic		
LC0238	TGGAACAAGGGCAGAAGA	7060–7077	512	Detection of *Tvsp* and *Bar* gene
LC0083	GAAGGCACGCAACGCCTACGA	7551–7571		
LC0220	ATGAGAAGGTGTGGGTATAGTTTG	9152–9175	945	Detection of *nos* and *Hs1pro^−1^* gene
LC0244	GCAAGACCGGCAACAGGA	10,079–10,096		
LC0236	CAGGTGGGTGTAGAGCGTG	7472–7490	1365	Detection of *2 × 35S* and *Bar* gene
LC0242	AGGAGGTTTCCGGATATTACC	8816–8836		

**Table 4 ijms-23-06849-t004:** Agronomic traits of T6 ZHs1-2 transformation event in the field.

Traits	T6 ZHs1-2	NT
Plant height(cm)	78.6 ± 0.9	78.6 ± 1.9
Height of bottom pod (cm)	8.6 ± 0.7	7.6 ± 1.2
Number of main stems	19.2 ± 0.7	19.6 ± 0.4
Number of branches	4.2 ± 0.7	4.6 ± 0.8
Number of pods per plant	61.8 ± 5.2	47.0 ± 6.1
Number of seeds per plant	155.0 ± 13.2	114.8 ± 17.4
Grain yield per plant	26.0 ± 2.2	19.1 ± 2.9
100-seed weight (g)	16.8 ± 0.1	16.7 ± 0.1
Grain yield (kg/hm^2^)	1733.5 ± 83.5	1495.4 ± 51.6

Data were collected from a T6 ZHs1-2 transformation event in a field trial in 2020 and 2021 in Shanxi province, China. NT, nontransgenic control ‘*Tianlong 1*’.

**Table 5 ijms-23-06849-t005:** T-DNA vector elements, location, size, and function in each plasmid vector.

T-DNA Vector Elements	Location in Vector	Size (bp)	Function
LB	6426–6451	26	T-DNA left border sequence of *Agrobacterium* C58, required for T-DNA transfer
Tvsp	6741–7292	552	Soybean storage protein terminator
*Bar*	7294–7869	564	Code PAT protein, relieves toxicity of glufosinate
TEV enhancer	7870–8014	145	5’ leader sequence of tobacco etch virus to enhance transcriptional level
2 × 35S	8015–8731	717	The tandem 35S promoter of cauliflower mosaic virus (CaMV)
CaMV 35S	8732–9145	414	The 35S promoter of cauliflower mosaic virus
*Hs1*	9152–10,000	849	Encodes sugar beet *Hs1^pro−1^* protein to improve the soybean cyst nematode resistance
nos	10,067–10,296	230	The terminator of the nopaline synthase gene to promote transcription termination
RB	10,693–10,717	26	T-DNA right border sequence of *Agrobacterium* C58, required for T-DNA transfer.

**Table 6 ijms-23-06849-t006:** Primer pairs for *Hs1^pro−1^*, *Bar* genes known sequences, and length of amplified fragments.

Target Gene Primers	Primer Sequence	Length of Amplified Fragments (bp)
*Bar*-F	CAGCTGCCAGAAACCCACGT	436
*Bar*-R	CTGCACCATCGTCAACCACT	
*Hs1*-1-F	GCTCTAGAATGAGAAGGTGTGGGTATAG	849
*Hs1*-1-R	GCTCTAGATCATTGTTTCGCAGTCCG	
*Hs1*-2-F	GGCACCATCCAAACTCGG	543
*Hs1*-2-R	CGAATAAGTGAGAGGATC	

**Table 7 ijms-23-06849-t007:** Name of primer, primer sequence (5′-3′), location in the vector (bp), amplified size (bp), and usage of Southern probes for checking *Hs1^pro−1^* and *Bar* genes.

Name of Primer	Primer Sequence (5′-3′)	Location in Vector (bp)	Amplified Size (bp)	Usage
LC0220	ATGAGAAGGTGTGGGTATAGTTTG	9152–9175	849	Check *Hs1^pro−1^* gene
LC0221	TCATTGTTTCGCAGTCCGATT	9980–10,000		
LC0236	CAGGTGGGTGTAGAGCGTG	7472–7490	231	Check *Bar* gene
LC0237	GTCAACTTCCGTACCGAGCC	7683–7702		

## Data Availability

All data supporting the conclusions of the present study have been documented in this article.

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
