# Peer review of "Generation and Identification of the Number of Copies of Exogenous Genes and the T-DNA Insertion Site in SCN-Resistance Transformation Event ZHs1-2"

_ijms, 2022, doi:10.3390/ijms23126849_

Round 1

Reviewer 1 Report

The manuscript lays emphasis on Identification of the number of copies of exogenous genes in Soybean with TDNA insertion. The manuscript is written well with a grey problem that the authors come up with.  The strengths and limitations of the work is well explored

What remains unclear is the T1/T2 steps as Figure 4 is not clear

There must be a pictorial methodology

The introduction first paragraph must be rewritten and paraphrased as some sections are matching with public text

The conclusions is missing

I made subtle edits in the attached document. kindly find it

Scores on a scale of 0-5 with 5 being the best

Language: 3.5

Novelty:3

Scope and relevance: 4

brevity: 3

Reviewer 2 Report

In the submitted manuscript by Tang et al., entitled “Generation, Identification of the number of copies of exogenous genes, and T-DNA insertion site in SCN resistance transformation event ZHs1-2” the authors generated overexpression of Hs1pro-1 gene by the Agrobacterium-mediated transformation in soybean, and identified their resistance to SCN infection. They also identified the gene copy number and T-DNA insertion sites in ZHs1-2 transformation event that has the least number of developed females per plant.

Overall, the interpretations seem reasonable. The manuscript writing is understandable and fairly easy to read, though there are a few instances where certain statements are ambiguous or not entirely correct. Grammatical mistakes in several instances make it difficult to understand what the authors want to convey.

Several grammatical, spelling errors and wrong sentence formation throughout the manuscript, I cannot point out all as there are no line numbers to indicate the errors.

The introduction seems very vague as the correlation and flow are missing between the sentences and paragraphs.

Fig. 1 Panel A, B, C &D can be moved as a supplementary figure.

Fig. 2: What do error bars represent, SE or SEM? How many replicates, transformation events, etc.? Please provide all the details in the figure legend.

Fig. 2: Explain the female index?

Many abbreviations, it will be informative to mention the expanded forms of NT, FI, etc.

Fig. 4: There are several instances where figure panels don’t correlate with the text. Panel ‘C’ is indicated in the figure legend but not mentioned in the figure? Similarly, panel “B’ is designated incorrectly in fig.

In the results section page 7 line 3 fig. 5C is mentioned but there is no fig. 5C.

Fig. 7: What do error bars represent, How many replicates? Relative expression to what, what’s the control? Please provide all the information in the figure legend.

Round 2

Reviewer 2 Report

Authors have addressed all my concerns and I have no further comments.